# Efficient Zero-Shot Coordination via Offline Policy Diversity and Online Belief Reasoning

## Abstract

A central challenge in multi-agent reinforcement learning is zero-shot coordination (ZSC): the ability of agents to collaborate with previously unseen partners. Existing approaches, such as population-based training or convention-avoidance methods, improve ZSC but typically rely on extensive online interaction, leading to high sample complexity. A natural alternative is to leverage preexisting interaction datasets through offline learning. However, offline training alone is insufficient for effective ZSC, as agents tend to overfit to the conventions present in the dataset and struggle to adapt to novel partners. To address this limitation, we propose an *offline-to-online ZSC* framework that combines offline dataset diversity with efficient online adaptation. In the offline stage, trajectories are embedded and clustered into behavioral modes to train specialized agents and their belief models, from which a best-response agent is learned. In the online stage, this agent is refined through belief-guided counterfactual rollouts, where belief models simulate alternative successor states under different teammate behaviors, thereby expanding the training distribution beyond the dataset. Experiments on the ZSC benchmark *Hanabi* in 2-player settings, as well as in human-AI coordination, demonstrate that our approach achieves state-of-the-art performance with unseen partners while significantly reducing the amount of online interaction.

## 1 Introduction

Multi-agent reinforcement learning (MARL) is a principled approach for studying how multiple agents can learn to coordinate in complex environments. The presence of multiple agents makes effective coordination a central challenge. In particular, agents must often cooperate with previously unseen partners—commonly referred to as *zero-shot coordination* (ZSC) (Hu et al., 2020a). Effective ZSC is essential in many real-world applications: autonomous vehicles must dynamically coordinate with unknown traffic participants; multirobot systems need to collaborate across heterogeneity to accomplish collective tasks; and human-AI teams must seamlessly align with novel users. More broadly, ZSC reflects the realities of open-agent ecosystems, where agents frequently encounter unfamiliar teammates. The challenge is amplified in *human-AI coordination*, as humans exhibit heterogeneous, context-dependent strategies and dynamically adapt to AI actions (Sycara et al., 2020; Li et al., 2020; 2021). Achieving effective ZSC therefore requires agents to generalize beyond narrow training conventions and reason about the latent strategy spaces of potential collaborators rather than relying solely on past interactions.

Recent work in ZSC has explored policy diversity (Liu et al., 2022) and convention-avoidance approaches (Hu et al., 2021; 2019; 2020a). These approaches typically rely on extensive online interaction or large-scale simulation, which can be costly or impractical in real-world domains. Moreover, many domains already have large offline interaction datasets and could serve as a foundation for robust coordination. While offline RL in multi-agent settings has been studied (Yang et al., 2021; Pan et al., 2022; Wang et al., 2023; Meng et al., 2023; Barde et al., 2024), offline ZSC remains largely unexplored. However, offline training alone has inherent limitations for ZSC: because it depends on the diversity and conventions present in the dataset, it can struggle to generalize to unseen partners and thus limits its effectiveness, as we will discuss in Sec. 4.4.

This limitation motivates us to investigate an *offline-to-online ZSC (off-to-on ZSC)* paradigm that leverages offline datasets to improve sampile efficiency while incorporating online fine-tuning to

achieve robust generalization to unseen partners. Our framework thus follows a two-phase structure: (i) offline training to exploit the diversity present in datasets, and (ii) online fine-tuning to adapt beyond the dataset and handle previously unseen partners. Specifically, in the *offline phase*, we embed trajectories into a latent space using a variational autoencoder and cluster them into distinct behavioral modes. Specialized agents are trained on each cluster with diversity-promoting objectives, and a best-response agent is then trained against this diverse pool to avoid overfitting to particular conventions. In the *online phase*, we fine-tune the best-response agent with belief-guided counterfactual rollouts, where pretrained belief models sample counterfactual successor trajectories under different teammate assumptions, thereby broadening the training distribution beyond the offline dataset. This combination of structured offline policy diversity and belief-driven online adaptation enables efficient and robust coordination with novel partners.

We validate our approach on the ZSC benchmark Hanabi in the 2-player setting and in human-AI coordination. In the offline-only regime, we show that agents overfit to dataset-specific conventions and fail to generalize, underscoring the need for online adaptation. Our offline-to-online pipeline achieves substantially higher cross-play performance and faster adaptation to unseen partners compared to both offline and online baselines, while also enabling more effective collaboration with human partners. Our main contributions are as follows:

• To the best of our knowledge, this is the first work to explicitly leverage offline datasets for zero-shot coordination.
• We propose an offline-to-online framework that combines population-based policy diversity in offline training with belief-guided adaptation in online fine-tuning, reducing reliance on training-time conventions and improving generalization to unseen partners.
• We conduct extensive evaluations on the ZSC benchmark *Hanabi*, as well as human-AI collaboration experiments, demonstrating that our approach achieves strong generalization to unseen partners with significantly improved sample efficiency.

## 2 BACKGROUND AND RELATED WORK

### 2.1 MULTI-AGENT REINFORCEMENT LEARNING

In this paper, we consider fully cooperative multi-agent reinforcement learning in partially observable environments, which is commonly modeled as *Decentralized Partially Observable Markov Decision Processes (Dec-POMDPs)* (Nair et al., 2003). Each agent $i$ selects actions according to its policy $\pi^i(a_t^i \mid \tau_t^i)$, conditioned on its action-observation history $\tau_t^i = (o_1^i, a_1^i, \ldots, o_t^i)$. After executing actions, agents receive a shared reward $r_t$ and their next observations, and this process repeats over time. The goal here is to learn a joint policy $\pi = (\pi^1, \ldots, \pi^n)$ that maximizes the expected cumulative reward. In multi-agent RL, several challenges naturally arise from the presence of multiple agents, such as non-stationarity (Papoudakis et al., 2019) and credit assignment under shared rewards (Foerster et al., 2018). Among these, an especially important problem is generalization to previously unseen partners, referred to as zero-shot coordination, where agents must successfully coordinate without prior joint training. Motivated by this, we focus on zero-shot coordination as a key challenge in cooperative multi-agent RL.

### 2.2 ZERO-SHOT COORDINATION

In cooperative MARL, as discussed above, the objective is to train a joint policy that maximizes the cumulative shared reward, which typically involves training and evaluating agents together as a team, i.e., *self-play* (SP). While SP can yield optimal joint performance during training, the learned policies often rely on arbitrary conventions that are not shared by independently trained partners. To address this, *Zero-Shot Coordination* (ZSC) (Hu et al., 2020a) has been studied, aiming to train agents that can coordinate effectively with novel partners without prior joint training. A common way to evaluate ZSC is through *cross-play* (XP), which formally measures the performance of independently trained policies when paired together at test time. Concretely, if $\pi_1$ and $\pi_2$ denote two joint policies trained independently, their XP performance is defined as

$$J_{\text{XP}}(\pi_1, \pi_2) = \tfrac{1}{2} \left( J(\pi_1^1, \pi_2^2) + J(\pi_2^1, \pi_1^2) \right), \tag{1}$$

where $\pi_i^1$ and $\pi_i^2$ denote the respective policies of agents 1 and 2. This formulation captures the requirement that policies trained independently must be compatible with one another, rather than

relying on arbitrary conventions established in self-play. Addressing this challenge has led to two main lines of research: *population-based training*, where agents are trained with a diverse set of partners for generalizable coordination, and *convention-avoidance methods*, which explicitly reduce agents' dependence on training-time conventions so that their learned policies generalize reliably to unseen partners.

**Population-based training** constructs a diverse population of partner policies and trains an agent as a common best response (BR) to this population. By explicitly promoting diversity among partners, PBT mitigates overfitting to specific conventions and improves coordination with unseen teammates. A representative example is the Trajectory Diversity method (Liu et al., 2022), which regularizes a population of policies to maximize the Jensen–Shannon divergence (JSD) between their trajectory distributions, thereby encouraging diverse behaviors. This diversity objective, combined with the training of a BR against the entire population, yields policies that achieve improved cross-play performance in ZSC settings.

**Convention-avoidance approaches** aim to mitigate the convention dependence of learning by grounding policies in explicit models of partner behavior. Instead of relying on arbitrary conventions formed during self-play, agents interpret observed actions under the assumption that they were generated by a simple baseline policy (often referred to as a level-0 policy). This grounding prevents convergence to incompatible strategies and enables consistent coordination with unseen partners. A representative method in this line of work is *Off-Belief Learning* (OBL) (Hu et al., 2021). OBL formalizes coordination as an inference problem: given an agent's action-observation history, the observed actions are assumed to have been produced by a baseline policy $\pi_0$, while the agent's own future behavior follows its learned policy $\pi_1$. The corresponding counterfactual value function is defined as

$$V^{\pi_0 \to \pi_1}(\tau^i) = \mathbb{E}_{\tau \sim B_{\pi_0}(\tau^i)} \left[ V^{\pi_1}(\tau) \right], \tag{2}$$

where $B_{\pi_0}(\tau^i)$ denotes the belief distribution over world states consistent with agent $i$'s history under $\pi_0$. By alternating between *belief inference* and *policy optimization*, OBL produces grounded and unique solutions that avoid reliance on arbitrary conventions and have demonstrated competitive ZSC performance in cooperative benchmarks such as Hanabi.

## 2.3 Offline Reinforcement Learning and Online Fine Tuning

Offline RL trains a policy to maximize expected returns from a fixed dataset without further environment interactions . A key challenge is extrapolation error, where limited coverage leads to inaccurate value estimates and degraded performance. To address this, single-agent offline RL often employs behavior-constrained objectives that regularize the policy toward the behavior policy, typically via an auxiliary behavior-cloning (BC) loss . In multi-agent settings, the difficulty is amplified: multiple agents expand the joint action space and induce non-stationarity, so fixed datasets rarely cover the full range of partner behaviors. Similar ideas have been applied in offline MARL, alleviating some of these issues but still falling short of robust generalization.

Particularly, this limitation becomes more severe in the context of ZSC. Even if offline MARL can mitigate extrapolation error during training, unseen partners at test time may exhibit behaviors absent from the dataset, leading to what can be viewed as a form of extrapolation error at execution. This makes it difficult to achieve reliable ZSC through offline learning alone. As we will demonstrate in Sec. 4.4, agents trained solely on fixed datasets tend to overfit to the conventions of their training partners. Thus, a central question is how to leverage offline data in a way that supports generalization to novel partners and this leads us to consider offline-to-online paradigm.

**Off-to-Online Learning** The offline-to-online paradigm combines the sample efficiency of offline pretraining with the adaptability of online fine-tuning. Prior work has shown its effectiveness in general multi-agent settings: for example, Multi-Agent Decision Transformers (MADT) (Meng et al., 2021) leverage offline datasets for subsequent online adaptation, while model-based methods such as MOTO (Rafailov et al., 2023) integrate predictive modeling with online rollouts to ensure a smooth transition. However, prior work *has not studied off-to-online learning in the context of ZSC*. This paper addresses this problem and demonstrates how combining offline pretraining with online adaptation can enable both efficiency and generalization in cooperative MARL.

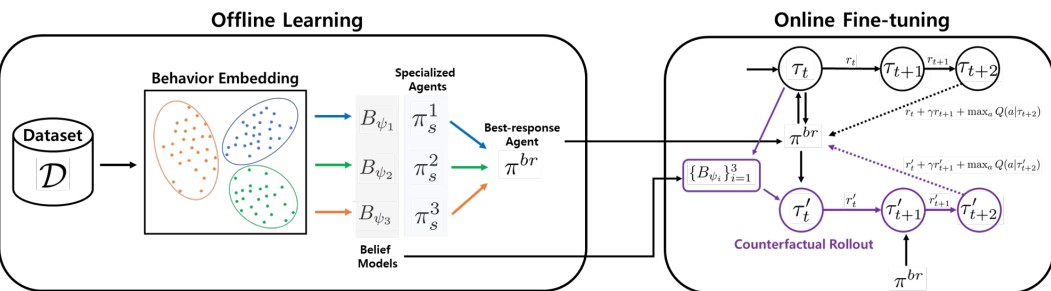

Figure 1: Overview of our offline-to-online framework for ZSC. Offline phase (left): trajectories from the dataset $\mathcal{D}$ are clustered into behavioral modes, from which specialized agents $\{\pi_s^i\}_{i=1}^3$ and their belief models $\{B_{\psi_i}\}_{i=1}^3$ are trained. A best-response agent $\pi^{br}$ is then bootstrapped against this diverse agent pool. Online phase (right): the BR agent is further adapted using belief-guided counterfactual rollouts, where belief models generate counterfactual successor states to construct enriched TD targets. This hybrid approach combines the efficiency of offline pretraining with the adaptability of online fine-tuning, enabling robust coordination with unseen partners.

## 3 METHODOLOGY

Zero-shot coordination has been actively studied, yet most existing approaches rely on extensive online interactions. This is because RL itself requires substantial environment experience, and in ZSC the need to train best-response agents against diverse partners further amplifies this demand. Leveraging pre-collected offline datasets therefore offers a promising way to improve sample efficiency, but how to effectively utilize such data for ZSC remains largely unexplored.

Our approach addresses this gap by combining two key principles of ZSC—*population diversity* to expose agents to a wide range of partner behaviors, and *belief grounding* to prevent reliance on arbitrary conventions—within an *offline-to-online learning framework*. Concretely, (i) in the **offline phase**, we extract diverse partner strategies from the dataset and train a best-response (BR) agent against them, and (ii) in the **online phase**, we fine-tune this BR agent using belief models that infer hidden teammate states and generate counterfactual trajectories. This framework combines the efficiency of offline pretraining with the adaptability of online fine-tuning, enabling robust generalization to unseen partners. An overview is illustrated in Fig. 1.

### 3.1 OFFLINE LEARNING: DIVERSE AGENT POOL AND BEST-RESPONSE AGENT

In the offline phase, the goal is to leverage the dataset to construct a pool of diverse coordination strategies and to train a best-response (BR) agent against them. We first learn trajectory representations and cluster them into distinct behavioral modes, from which specialized agents are trained to form a *Diverse Agent Pool* $\{\pi^1, \pi^2, \ldots, \pi^{k^*}\}$. The BR agent is then trained against this fixed pool, encouraging robustness to a wide range of partner behaviors (Osborne & Rubinstein, 1994).

#### 3.1.1 TRAINING A DIVERSE AGENT POOL

To construct a diverse set of coordination strategies from the offline dataset, we follow a three-step procedure: trajectory representation learning, behavioral clustering, and specialized agent training.

**Trajectory Representation and Clustering.** To identify diverse coordination strategies from the offline dataset, we first train a trajectory VAE (Lu et al., 2019; Gao et al., 2022; Yao et al., 2020) that encodes full trajectories $\tau = \{(o_t, a_t, r_t)\}_{t=1}^T$ into compact latent embeddings $z \in \mathbb{R}^d$. The VAE is trained with a standard objective that balances trajectory reconstruction and KL regularization (details in Appendix A.2). We then cluster the learned embeddings $\{z_j\}_{j=1}^N$ using adaptive $k$-means with silhouette analysis, yielding distinct behavioral modes that serve as the basis for training specialized agents.

**Specialized Policy and Belief Training.** Let $\mathcal{D}_i \subset \mathcal{D}$ denote the subset of trajectories assigned to cluster $i$ after clustering. Each specialized agent $\pi_s^i$ parameterized by $\theta_s^i$ is trained using data from $\mathcal{D}_i$. In order to train the specialized agent in a stable and diverse manner, inspired by behavior-

constrained approaches in offline RL and population-based training, we incorporate a BC loss to constrain policies toward the data distribution and a Jensen–Shannon divergence (JSD) regularizer to enforce diversity across agents, in addition to an RL term given by the $n$-step TD loss. The corresponding objective can be written as

$$\mathcal{L}(\theta_s^i) = \mathbb{E}_{\tau \sim \mathcal{D}_i}\Big[\underbrace{\big(R_t^n + \gamma^n \max_a \hat{Q}^i(\tau_{t+n}, a) - Q^i(\tau_t, a_t)\big)^2}_{\text{TD loss}} + \lambda_{BC}\underbrace{\text{CE}(\pi_s^i(\cdot|\tau_t), a_t)}_{\text{BC loss}}$$

$$-\lambda_{JSD}\underbrace{\frac{1}{k^*}\sum_{j=1}^{k^*}[\text{KL}(\pi_s^j(\cdot|\tau)\|\pi_s^i(\cdot|\tau))]}_{\text{JSD regularizer}}\Big]. \tag{3}$$

Here, $R_t^n = \sum_{k=0}^{n-1}\gamma^k r_{t+k}$ is the $n$-step return, $Q^i$ and $\hat{Q}^i$ denote the current and target critics of agent $i$, $a_t$ is the dataset action, $\lambda_{BC}, \lambda_{JSD}$ are weighting coefficients, and $k^*$ is the number of clusters. This procedure yields a *Diverse Agent Pool* $\{\pi_s^1, \cdots, \pi_s^{k^*}\}$ that captures complementary strategies while avoiding collapse into narrow conventions.

We additionally train a belief model $B_{\psi_i}$ for each specialized agent $\pi_s^i$, which will later be used in online fine-tuning. Following Hu et al. (2020b; 2021), each belief model is trained in a supervised manner to infer latent teammate states (e.g., hidden hands in Hanabi) from action–observation histories. We implement $B_{\psi_i}$ as an encoder–decoder network and optimize it by minimizing the negative log-likelihood of the ground-truth hidden state. This yields belief models that can generate counterfactual states, enabling counterfactual rollouts during online adaptation. Full architectural details and the loss function are provided in Appendix A.3.

### 3.1.2 TRAINING THE OFFLINE BEST RESPONSE AGENT

After constructing the diverse agent pool, we train a best-response agent that approximates responses to all pool members. In the offline setting this is challenging, since evaluating best responses typically requires direct interaction with the pool. To address this, we bootstrap from the value functions of specialized agents: the BR agent computes TD targets using their critics, while its own policy is regularized with behavior cloning for stability. The objective of the BR agent, parameterized by $\theta_{br}$, is given by

$$\sum_{i=1}^{k*}\mathbb{E}_{\tau \sim \mathcal{D}_i}\left[\big(R_t^n + \gamma^n \max_a Q^i(\tau_{t+n}, a) - Q^{br}(\tau_t, a_t)\big)^2 + \lambda_{BC}\,\text{CE}(\pi^{br}(\cdot|\tau_t), a_t)\right]. \tag{4}$$

Here, $Q^{br}$ and $\pi^{br}$ denote the critic and policy of the BR agent, respectively, and $Q^i$ is the critic of specialized agent $i$. This allows the BR agent to approximate best responses without direct interaction, leveraging the specialized agents' value functions as surrogates for diverse partner strategies.

### 3.2 ONLINE FINE-TUNING VIA BELIEF-BASED COUNTERFACTUAL ROLLOUTS

In the offline phase, we constructed a diverse pool of specialized agents and their corresponding belief models, together with a BR agent trained against this pool. While this procedure improves over training a single agent directly on the dataset, the BR agent remains tied to the conventions present in the dataset and thus struggles to generalize to unseen partners. In particular, offline BR training alone cannot escape overfitting to training-time conventions, motivating the need for an online adaptation mechanism.

To this end, we introduce an online fine-tuning stage that leverages the belief models trained in the offline stage to fine-tune the BR agent. These models generate counterfactual successor trajectories. That is, we generate $M$ counterfactual successor trajectories $\{\tau'_{t+1,m}\}_{m=1}^M \sim B_\psi^{(i)}(\tau_{t+1} \mid \tau_t^{(i)})$ by sampling from the belief model of specialized agents. Intuitively, these counterfactual successor trajectories correspond to hypothetical continuations obtained by assuming that, instead of the actual partner, one of the specialized agents from the offline pool had acted as the teammate at this step. This allows the BR agent to update its value estimates under a broader distribution of plausible partner behaviors, thereby mitigating overfitting to training-time conventions and equipping

the agent with broader experience that facilitates more effective adaptation to unseen teammates. These counterfactual samples are then used to compute the belief-guided target value for $n$-step TD learning:

$$y_t^{\text{CF}} = R_t^n + \gamma^n \frac{1}{M} \sum_{m=1}^{M} \max_{a_{t+n}} \hat{Q}^{br}(\tau'_{t+n,m}, a_{t+n}) \tag{5}$$

where $\hat{Q}^{br}$ denotes the target critic of the BR agent. The BR agent is then updated by minimizing the counterfactual TD loss:

$$\mathcal{L}(\theta_{br}) = \mathbb{E}\Big[\big(y_t^{\text{CF}} - Q^{br}(\tau_t, a_t)\big)^2\Big]. \tag{6}$$

This training procedure allows the BR agent to update its value estimates using belief-guided successor trajectories rather than relying solely on on-policy samples. In practice, this methodology enables adaptation to new teammates by (i) exposing the agent to a broader distribution of plausible transitions, (ii) reducing dependence on the support of the offline dataset, and (iii) incorporating latent information inferred by belief models into the critic's updates.

## 4 EXPERIMENTS

### 4.1 HANABI GAME

We use the cooperative card game *Hanabi* as our primary benchmark. Hanabi is a fully cooperative, partially observable card game widely used for ZSC research (Bard et al., 2020) and is considered one of the most challenging environments for cooperative multi-agent learning, combining strict partial observability, limited communication, and the need for long-term coordination. The game uses a 50-card deck across five colors and ranks 1–5; the team's score equals the number of correctly played cards (maximum 25). Each player sees only their partner's hand, not their own, and communication is bottlenecked by a shared pool of eight hint tokens and three life tokens. On each turn, a player may *play*, *discard*, or spend a *hint* token to indicate all cards of a chosen color or rank in their partner's hand; incorrect plays lose a life, and discards or completing a stack with a 5 restore a hint token. The game ends when all life tokens are lost, the deck is exhausted, or all stacks are completed. Beyond these mechanics, Hanabi is particularly challenging because players must decide not only which hints to provide, but also interpret the intentions behind received hints and remember them across turns. Effective coordination thus hinges on modeling partners and establishing implicit conventions, closely reflecting the desiderata of ZSC, where agents must generalize and collaborate effectively with previously unseen teammates. In our experiments, we evaluate the two-player variant of Hanabi in both simulation and human experiments.

### 4.2 TRAINING DETAILS

We train agents offline using a 3-step TD loss combined with a behavior cloning loss, with weight $\lambda_{BC} = 0.4$. Both medium- and expert-replay datasets are used for training. Following Hu et al. (2021), we adopt a recurrent Q-learning backbone based on R2D2 with an LSTM hidden size of 512. Detailed hyperparameters are provided in Appendix A.4

### 4.3 OFFLINE DATASET

To evaluate our approach under controlled conditions, we construct synthetic offline datasets using the open-sourced OBL Hu et al. (2021) implementation. We generate two types of datasets that differ in quality: *Medium-Replay* and *Expert-Replay*. Each dataset is obtained by saving the replay buffer of policies trained under medium- or expert-level settings, respectively. For both settings, we collect data from 12 independent training seeds, resulting in approximately 200k gameplay episodes per dataset. The Medium-Replay dataset achieves an average score of $17.05 \pm 3.24$, reflecting substantial variability and the presence of suboptimal coordination behaviors. In contrast, the Expert-Replay dataset achieves a higher average score of $23.41 \pm 3.31$, with trajectories that exhibit more refined strategies and stronger coordination. These two datasets thus provide complementary conditions: the Medium-Replay dataset highlights challenges in learning from noisy, imperfect conventions, while the Expert-Replay dataset represents higher-quality but narrower coordination strategies.

## 4.4 OFFLINE ZSC: CHALLENGES AND LIMITATIONS

ZSC aims to enable agents to work effectively with previously unseen partners, but this is challenging since in practice each agent is trained from different experiences, such as variations in random seeds or training data, which often lead to the emergence of distinct conventions. When paired with a new partner that follows unfamiliar conventions, an agent is unable to correctly interpret their observations or actions, resulting in coordination failure. This problem is exacerbated in the offline setting. When agents are trained from fixed datasets, their learned policies become tied to the conventions represented in those datasets. Even if a dataset is relatively diverse, agents still rely heavily on conventions specific to the trajectories it contains, limiting their ability to generalize to unseen strategies. Consequently, offline training alone is insufficient for effective ZSC and highlights the necessity of online fine-tuning to adapt to novel partners.

We illustrate this limitation empirically by evaluating agents trained on different offline datasets. Fig. 2 compares the self-play and cross-play performance of agents trained on

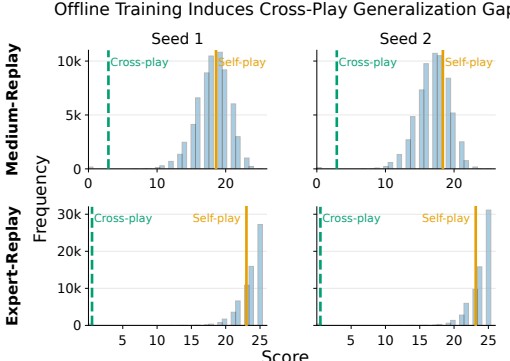

Figure 2: Offline training induces a cross-play generalization gap. Results on two Medium-Replay datasets (top) and two Expert-Replay datasets (bottom) show return distributions, with self-play performance (orange solid) remaining high but cross-play performance (green dotted) dropping sharply. This indicates that agents overfit to dataset-specific conventions, leading to poor zero-shot coordination.

pairs of Medium-Replay datasets (first row) and Expert-Replay datasets (second row). In each case, the two datasets exhibit different return distributions, and agents trained on one dataset achieve strong SP performance when paired with replicas of themselves. However, when paired with agents trained on the other dataset, XP performance drops sharply. This shows that agents internalize dataset-specific conventions during offline training: they coordinate well with partners exposed to the same dataset but fail to align with partners trained on different datasets. Consequently, even reasonably diverse datasets are insufficient for robust zero-shot coordination, underscoring the need for online adaptation to bridge these convention gaps.

## 4.5 NUMERICAL RESULTS

To evaluate the efficacy of our proposed pipeline, we compare it against state-of-the-art baselines and perform ablation studies to isolate the contribution of each component.

**Baselines:** We compare our proposed pipeline against several baseline methods and ablation variants. The baseline methods include (a) SAD (Hu & Foerster, 2019), a strong offline multi-agent learning approach designed for both self-play and cross-play; (b) OBL (Foerster et al., 2019), a leading zero-shot coordination method that employs online adaptation via belief modeling.

**Evaluation Metrics:** We evaluate all methods in terms of their self-play and cross-play performance measured at different training epochs. Self-play performance is quantified as the average score achieved by an agent when paired with itself, reflecting the efficiency and convergence of training. Cross-play performance is evaluated under *seed variation*, where we measure the average score across agents trained with different random seeds, following standard evaluation protocols for zero-shot coordination Hu et al. (2020a; 2021).

Fig. 3 presents results on the 2-player Hanabi benchmark, where the offline stage is trained with only 200k episodes—equivalent to a single epoch of online experience. This highlights the sample efficiency of our offline-to-online pipeline.

In terms of cross-play, the key metric for ZSC, our approach achieves strong initialization from offline learning, starting at substantially higher scores than baselines, especially when trained on the expert-replay dataset. During online adaptation, our method converges faster and more stably than

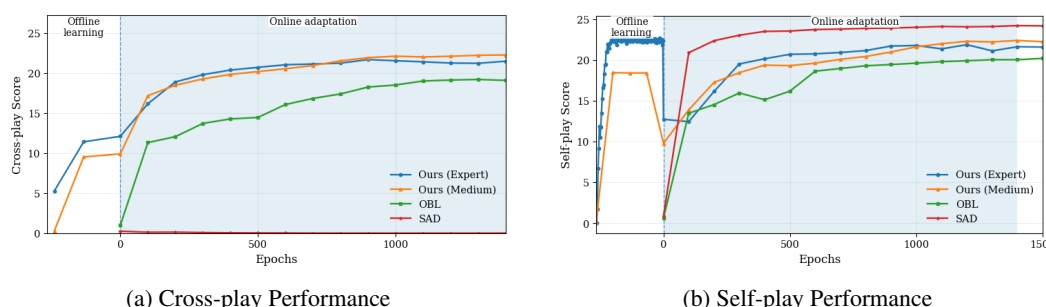

(a) Cross-play Performance          (b) Self-play Performance

Figure 3: Results on the 2-player Hanabi benchmark: (a) Cross-play performance and (b) self-play performance over training epochs. The dashed vertical line at epoch 0 separates the offline learning stage from the online adaptation stage. Our method is shown with Expert-Replay data (blue) and Medium-Replay data (orange), compared against OBL (green) and SAD (red).

Table 1: Comparison of different strategies on self-play and cross-play performance for 2 player.

| Strategy | Self-Play Score ↑ | Cross-Play Score ↑ | # Train Samples ↓ | |
|---|---|---|---|---|
| | | | Offline | Online |
| MARL-BC | $17.92 \pm 0.25$ | $9.58 \pm 1.47$ | 200k | - |
| OBR-C | $17.08 \pm 0.42$ | $9.92 \pm 1.15$ | 200k | - |
| OBR-Ours | $17.49 \pm 0.36$ | $10.56 \pm 0.84$ | 200k | - |
| SAD | $24.19 \pm 0.02$ | $2.21 \pm 0.22$ | - | 96.064M |
| OBL (level-1) | $20.92 \pm 0.07$ | $20.85 \pm 0.03$ | - | 142.35M |
| OBL (level-2) | $23.41 \pm 0.03$ | $23.24 \pm 0.03$ | - | 230.86M |
| OBL (level-3) | $23.93 \pm 0.01$ | $23.68 \pm 0.05$ | - | 356.76M |
| OBL (level-4) | $24.10 \pm 0.01$ | $23.76 \pm 0.06$ | - | 425.33M |
| Ours-wC | $23.01 \pm 0.0$ | $23.61 \pm 0.05$ | 200k | 256.02M |
| Ours | $22.84 \pm 0.12$ | $23.65 \pm 0.13$ | 200k | 229.38M |

both OBL and SAD, demonstrating its advantage in generalization to unseen partners. On self-play, we observe a temporary performance drop when online learning begins. This effect is expected, as belief-guided counterfactual rollouts encourage the agent to deviate from dataset-specific conventions in order to generalize. Notably, while SAD attains the highest self-play scores overall, its cross-play performance remains very poor, underscoring its reliance on narrow conventions rather than robust zero-shot coordination.

## 4.6 EMPIRICAL ANALYSIS

**Offline Phase Analysis.** We first analyze the effectiveness of the offline learning stage before online fine-tuning by comparing three variants: (a) a naive baseline using offline MARL with behavior cloning (MARL-BC), trained on the entire dataset, which represents a standard approach without explicit treatment of conventions; (b) an offline best-response agent trained directly on the entire dataset without trajectory clustering or diversity regularization (denoted as OBR-C); and (c) our method prior to online adaptation, where the best-response agent is trained with a Diverse Agent Pool constructed from clustered trajectories (denoted as OBR-Ours). This comparison isolates the role of structured offline diversity: if effective, the Diverse Agent Pool should yield stronger cross-play performance than both the naive offline baseline and the BR variant without diversity, highlighting the importance of explicitly addressing conventions even in the offline phase. Table. 1 shows that OBR-Ours achieves a higher cross-play score (10.56) than both MARL-BC (9.58) and OBR-C (9.92), while maintaining similar self-play performance. This indicates that incorporating structured diversity in the offline phase provides a measurable benefit, though overall cross-play performance remains limited without online adaptation.

**Online Phase Analysis.** We next analyze the role of belief-guided counterfactual rollouts during online adaptation. To this end, we compare two variants: (a) Ours w/o Counterfactual (Ours-wC), which uses offline BR training followed by standard online fine-tuning without counterfactual roll-outs; and (b) Ours, the complete pipeline that incorporates both offline BR training and belief-guided counterfactual rollouts. This comparison isolates the effect of counterfactual reasoning in online learning. If effective, the belief-guided variant should achieve higher cross-play performance with unseen partners, as it enables the agent to adapt beyond the conventions present in the offline dataset and generalize more efficiently to novel teammates. Table. 1 shows that both variants achieve strong cross-play performance, but the belief-guided version (Ours) attains slightly better generalization while also requiring fewer online samples. This indicates that counterfactual rollouts provide a consistent benefit in sample efficiency and adaptation to unseen partners.

### 4.7 HUMAN-AI COORDINATION

Beyond coordination with AI agents, we also evaluate whether our approach extends to collaboration with human partners. We recruited 10 members of a local board game club, none of whom were familiar with Hanabi. Each participant played one game with two AI partners, presented in random order: our agent, the state-of-the-art agent OBL, and SAD. To control for variance due to deck order, which can strongly affect Hanabi outcomes, we reused the same seeds across all conditions.

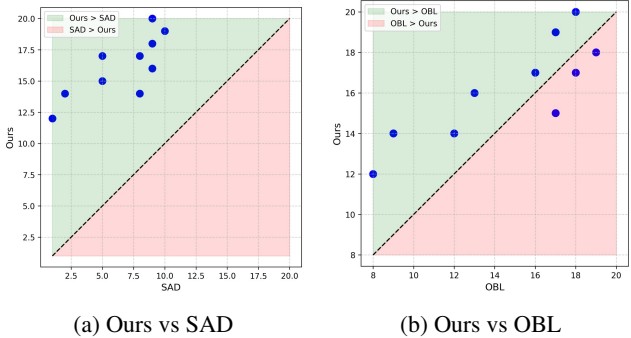

(a) Ours vs SAD    (b) Ours vs OBL

Figure 4: Comparison of human collaboration scores with BR agent against SAD and OBL.

Humans paired with our agent achieved an average score of 16.20, compared to 14.70 with OBL and 6.60 with SAD. These results show a clear trend: participants consistently achieved higher scores when paired with our agent than with either OBL or SAD. While the study is limited in scale, this trend suggests that our approach enables AI to align more effectively with human strategies in cooperative zero-shot settings, highlighting its potential for human-AI collaboration.

## 5 CONCLUSION

We present a novel offline-to-online multi-agent learning pipeline for zero-shot coordination that combines offline training with online fine-tuning. During offline training, we construct a Diverse Agent Pool through trajectory clustering and train a best-response agent against this pool to promote diversity and robustness from pre-collected data. During online fine-tuning, we introduce belief-guided counterfactual rollouts that leverage pretrained belief models to generate counterfactual transitions, enabling efficient adaptation to unseen partners. Experiments in Hanabi show that this pipeline achieves state-of-the-art performance in both seed and dataset variation settings, with notable improvements in sample efficiency. Furthermore, our method achieves strong human-AI collaboration performance, demonstrating robust generalization to human partners. These results highlight the promise of our approach as a foundation for practical, adaptive multi-agent systems that leverage preexisting datasets.

**Limitaion** Our approach fundamentally depends on the quality and coverage of the offline dataset, which may limit the diversity and generalization of the learned agents. Moreover, specialized agent training and counterfactual belief–based best response learning introduce additional hyperparameters that require tuning, and the assumption of accurately learnable belief models may break down in domains with highly complex hidden states.

**Future work** Future work includes mitigating the reliance on offline datasets, for example through data augmentation or hybrid collection strategies, and extending the pipeline to larger-scale multi-agent systems such as real-time strategy games or robotics, where scalability and coordination become more challenging.

**Ethics Statement**  Our research involves human-subject experiments through voluntary participation in gameplay studies with the cooperative card game Hanabi. All participants provided informed consent prior to involvement, and we collected no personally identifiable or sensitive information. We adhered strictly to the ICLR Code of Ethics to ensure responsible stewardship of research, minimizing any foreseeable risks to participants. Our methodology respects privacy, fairness, and scientific integrity, with no use of data beyond approved purposes and a commitment to transparency and reproducibility. The study posed no harm to participants or the environment. Any use of pretrained models and data complies with ethical standards, and all contributions have been acknowledged accordingly.

**Reproducibility Statement**  To ensure reproducibility of our work, we provide comprehensive details across the main text, appendix, and supplemental materials. Our novel offline-to-online zero-shot coordination framework is described with clear algorithmic explanations. The appendix includes detailed proofs of key theoretical claims and assumptions. We provide access to an anonymous downloadable source code repository as supplementary material to facilitate replication of our models and experiments. Additionally, complete descriptions of datasets used, data processing steps, and hyperparameters are included in the supplementary materials. These efforts enable full reproduction of our experimental results and theoretical findings.

**LLM Usage Statement**  Large Language Models (LLMs) were used solely as a general-purpose assistive tool for grammar correction and writing refinement during the preparation of this manuscript. LLMs did not contribute to the research ideation, methodology, experimental design, or interpretation of results. The authors take full responsibility for all content presented in this paper, including any text generated or polished with the help of LLMs. No part of the paper was generated independently by LLMs, and all scientific claims, data, and conclusions are the authors' original work.

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

# A   APPENDIX

## A.1   CODE

For reproducibility, we provide our code at `https://anonymous.4open.science/r/offline-zsc-br-1174`.

## A.2   CLUSTERING: TRAJECTORY REPRESENTATION LEARNING

Given the offline dataset $\mathcal{D}$, we learn compact representations using a TrajectoryVAE. This approach addresses the fundamental challenge of modeling multi-modal trajectory distributions arising from different strategic behaviors in cooperative settings (Zhao et al., 2024a; Zhu et al., 2024). For example, in Hanabi, some agents exhibit an *immediate play strategy*, playing cards immediately after receiving positive hints, while others follow a *conservative confirmation strategy*, waiting for additional information before acting. These behavioral differences create distinct trajectory patterns in state-action-reward sequences that TrajectoryVAE can capture and separate.

$$\mathcal{L}_{VAE} = \mathbb{E}_{\tau \sim \mathcal{D}} \left[ \sum_{t=1}^{T} \mathcal{L}_{recon}(a_t, r_t | \hat{a}_t, \hat{r}_t) + \beta \cdot \text{KL}(q_\phi(\mathbf{z}|\tau) \| \mathcal{N}(0, \mathbf{I})) \right] \tag{7}$$

TrajectoryVAE learns to encode full trajectory sequences $\tau = \{(o_t, a_t, r_t)\}_{t=1}^{T}$ into low-dimensional latent representations $\mathbf{z} \in \mathbb{R}^d$ that capture the underlying behavioral patterns. The encoder network $q_\phi(\mathbf{z}|\tau)$ processes temporal sequences of observations, actions, and rewards through recurrent layers to produce distributional parameters $\{\boldsymbol{\mu}_\phi(\tau), \boldsymbol{\sigma}_\phi(\tau)\}$ of a multivariate Gaussian posterior, while the decoder network $p_\theta(\hat{a}_t, \hat{r}_t | \mathbf{z}, o_{<t})$ takes both the latent representation $\mathbf{z}$ and previous observations $o_{<t}$ as input to reconstruct actions $\hat{a}_t$ and rewards $\hat{r}_t$ using the reparameterization trick $\mathbf{z} = \boldsymbol{\mu}_\phi(\tau) + \boldsymbol{\sigma}_\phi(\tau) \odot \boldsymbol{\epsilon}$ where $\boldsymbol{\epsilon} \sim \mathcal{N}(0, \mathbf{I})$ (Kingma & Welling, 2014). The training objective in 7 combines reconstruction accuracy with KL regularization, where $\mathcal{L}_{recon}$ measures the fidelity between ground truth trajectory elements $\{a_t, r_t\}$ and their reconstructed counterparts $\{\hat{a}_t, \hat{r}_t\}$ using cross-entropy loss for discrete actions and mean squared error for rewards in continuous space, while the KL term enforces that the learned posterior remains close to a standard Gaussian prior, promoting regularity in the latent space and enabling meaningful interpolation between behavioral modes (Higgins et al., 2017; Venkatraman, 2023).

The learned trajectory embeddings $\{\mathbf{z}_j\}_{j=1}^{N}$ are clustered to identify distinct behavioral modes within the dataset. We employ **k-means clustering with adaptive cluster selection** based on the silhouette score (Rousseeuw, 1987), which measures the quality of clustering by evaluating both intra-cluster cohesion and inter-cluster separation.

The optimal number of clusters $k^*$ is determined by maximizing the average silhouette score:

$$k^* = \underset{k \in [2, K_{max}]}{\arg\max} \frac{1}{N} \sum_{i=1}^{N} S_i(k) \tag{8}$$

where $S_i(k)$ represents the silhouette score for the trajectory $i$ in the clusters $k$ (Zhao et al., 2024b). This adaptive approach ensures that the clusters discovered correspond to meaningful behavioral differences rather than arbitrarily partitioning the data set (Wang et al., 2025). Each resulting cluster $C_k$ represents a collection of trajectories exhibiting similar strategic behaviors, enabling specialized policy training on homogeneous behavioral subsets.

## A.3   BELIEF MODEL LEARNING

Following Hu et al. (2020b), we train each belief model $B_\psi^{(i)}$ to predict the hidden hand of agent $i$ in Hanabi. Similar to Hu et al. (2021), given agent $i$'s action-observation history $\tau_t^{(i)}$, the model outputs an auto-regressive distribution over the $n$ cards in the agent's hand:

$$p\big(h_{1:n}^{(i)} \mid \tau_t^{(i)}\big) = \prod_{k=1}^{n} B_\psi^{(i)}\big(h_k^{(i)} \mid \tau_t^{(i)}, h_{1:k-1}^{(i)}\big). \tag{9}$$

We implement $B_\psi^{(i)}$ with an RNN encoder that processes $\tau_t^{(i)}$ into a hidden state that summarizes the history, followed by an RNN decoder initialized from this state that emits a card distribution per step $k$, conditioning on previous predictions $h_{1:k-1}^{(i)}$. The belief network is trained via supervised learning by minimizing the negative log-likelihood of the true hand:

$$\mathcal{L}_{\text{belief}}^{(i)}(\psi) = -\sum_{k=1}^{n} \log B_\psi^{(i)}\big(h_k^{(i)} \mid \tau_t^{(i)}, h_{1:k-1}^{(i)}\big) \tag{10}$$

This loss encourages accurate auto-regressive hand reconstruction and enables sampling of plausible hidden hands for counterfactual return estimation during online adaptation.

## A.4 HYPERPAREMETER

Table 2: Hyper-parameters for the offline training

| Hyper-parameters | Value |
|---|---|
| # dataset | |
|   dataset_size | 200,000 trajectories |
|   max_trajectory_length | 80 |
| # optimization | |
|   optimizer | Adam (Kingma & Ba, 2015) |
|   lr | 5e-4 |
|   eps | 1.5e-5 |
|   grad_clip | 5 |
|   batchsize (coop agent) | 128 |
|   batchsize (best response agent) | 256 |
| # Q learning | |
|   n_step | 3 |
|   discount_factor | 0.999 |
|   target_network_sync_interval | 1000 |
| # architecture | |
|   rnn_hid_dim | 512 |
|   num_lstm_layer | 2 |
| # losses | |
|   BC_loss_coeff ($\lambda_{bc}$) | 0.4 |
|   JSD_loss_coeff ($\lambda_{jsd}$) | 0.1 |
|   CQL_loss | Not used |

Table 3: Hyper-parameters for the online adaptation paradigm

| Hyper-parameters | Value |
|---|---|
| # replay buffer related | |
| burn_in_frames | 10,000 |
| replay_buffer_size | 262,144 |
| max_trajectory_length | 80 |
| # optimization | |
| optimizer | Adam (Kingma & Ba, 2015) |
| lr | 6.25e-05 |
| eps | 1.5e-05 |
| grad_clip | 10 |
| batchsize | 128 |
| # Q learning | |
| n_step | 1 (belief-based), 3 (non-belief based) |
| discount_factor | 0.999 |
| target_network_sync_interval | 2500 |
| exploration $\epsilon$ | $\epsilon_0 \dots \epsilon_n$, where $\epsilon_i = 0.1^{1+7i/(n-1)}$, $n = 80$ |
| num_agents | 3 |
| # cooperative agent updates | |
| update_coop_agents | True |
| update_coop_agents_freq | 50 |
| update_coop_agents_belief | True |
| update_coop_agents_belief_freq | 50 |
| coop_agent_belief_sync_freq | 5000 |
| # architecture | |
| rnn_hid_dim | 512 |
| num_lstm_layer | 2 |
| fc_only | 0 |

