# OpenReview forum: "Efficient Zero-Shot Coordination via Offline Policy Diversity and Online Belief Reasoning"
_ICLR.cc/2026/Conference — ICLR 2026 Conference Desk Rejected Submission_

### Official Review · Reviewer_1FBQ · 2025-10-26

**Soundness:** 2
**Presentation:** 3
**Contribution:** 2
**Rating:** 4
**Confidence:** 4

**Summary:**

The paper addresses the challenge of zero-shot coordination (ZSC) in multi-agent reinforcement learning (MARL), where agents must cooperate with previously unseen partners without additional training. While traditional approaches rely heavily on online RL and environment interactions, this work proposes a offline-to-online ZSC framework, which improves sample efficiency and generalization. The framework is built on two key components: Offline Policy Diversity which leverage's existing offline datasets to train diverse populations and then training a offline best response agent, and Online Belief Reasoning, which uses belief model to generate counterfactual rollouts and update model. Through the combination of offline behavior diversity and belief-based reasoning, the method achieves substantial gains in zero-shot coordination while maintaining high sample efficiency. Experiments on the Hanabi benchmark and human-AI collaboration studies demonstrate that the proposed approach outperforms baseline methods.

**Strengths:**

1. The paper is well-written and clearly structured, making it easy to follow.
2. The integration of offline reinforcement learning (RL) techniques into zero-shot coordination (ZSC) makes sense, effectively improving sample efficiency and reducing training costs. The online fine-tuning part, which leverages a belief model to generate counterfactual rollouts for model adaptation, is well motivated.
3. The discussion on experiments is comprehensive. Moreover, the inclusion of human-AI coordination experiments demonstrates the practical applicability of the proposed method.
4. The authors provide open-source code, ensuring the reproducibility of the proposed approach.

**Weaknesses:**

1. The use of offline RL in zero-shot coordination (ZSC) appears somewhat trivial. I think numerous prior researchers have already explored offline methods for ZSC problems, such as CooT: Learning to Coordinate In-Context with Coordination Transformers (arXiv:2506.23549). The key challenge remains that offline training often struggles to outperform online ZSC approaches. As demonstrated in this paper’s experimental results, the proposed method similarly does not achieve a substantial performance improvement.
2. The set of baseline methods used for comparison is relatively limited and somewhat outdated. More recent population-based approaches, such as MEP, TrajeDi, and COLE, should be included to strengthen the evaluation and provide a more comprehensive comparison.
3. The experimental environment is relatively narrow. Although Hanabi is used in some ZSC studies, evaluation on more contemporary benchmarks—such as Overcooked—would provide stronger evidence of the method’s generalizability and practical effectiveness.
4. The number of participants in the human-AI coordination experiments is relatively small, and the evaluation metrics are limited, raising concerns about potential randomness and the reliability of the reported findings.

**Questions:**

1. The experimental results are not entirely convincing. As mentioned in the Weaknesses, the performance improvement is limited, and the small number of outdated baselines within a single environment weakens the overall empirical evidence.
2. According to my understanding, the offline dataset is collected using models trained by previous methods (e.g., OBL). Has the training cost of generating this dataset been accounted for? As noted in the paper, the quality and coverage of the offline dataset significantly affect final performance. It would be valuable to discuss whether there are more efficient or effective approaches to constructing such offline datasets.
3. In Figure 3, the blue background region appears misaligned and the figures are visually unpolished. Additionally, in Table 1, the best-performing results should be highlighted more clearly to improve readability and presentation quality.

---

### Official Review · Reviewer_7HYh · 2025-10-29

**Soundness:** 3
**Presentation:** 2
**Contribution:** 1
**Rating:** 2
**Confidence:** 5

**Summary:**

The paper addresses the challenge of Zero-Shot Coordination (ZSC), where agents must collaborate with unseen partners, a task often hampered by the high sample complexity of traditional online methods. The authors propose an offline-to-online ZSC framework to leverage existing interaction datasets and improve efficiency. In the offline phase, they cluster trajectories into distinct behavioral modes to train a Diverse Agent Pool and their corresponding belief models, subsequently training a best-response agent against this pool. In the online phase, they fine-tune this best-response agent using belief-guided counterfactual rollouts to generate hypothetical successor states, thereby expanding the agent's experience beyond the conventions of the original dataset. The authors evaluate their method on the Hanabi benchmark and found that their approach achieved strong cross-play performance and faster adaptation to novel partners compared to several online baselines, while also demonstrating effective collaboration with human partners.

**Strengths:**

- The proposed offline-to-online pipeline achieves strong cross-play performance and converges faster during online adaptation compared to online baselines, demonstrating improved sample efficiency.
- The ablation studies confirm that the clustering of behavioral modes and training a Best-Response agent against this diverse pool provides a measurable and stable initialization for ZSC
- The method shows positive preliminary results for human-AI coordination

**Weaknesses:**

- The core claim that this is the first work leveraging offline data for ZSC is demonstrably incorrect. The related work section is sparse, critically omitting recent, highly relevant concurrent efforts (e.g., GAMMA, GOAT) which already use offline data to learn latent partner embeddings for use in online RL. Furthermore, it is missing essential citations for Population-Based Training (Comedei, MEP, ROTATE, Trajedei), Zero-Shot Coordination (E3T, CEC), and general offline-to-online learning (e.g., RLPD).
- The approach used for generating the agent pool appears highly similar to existing work like GAMMA, with the primary technical addition being the clustering of the latent vectors. This similarity further undermines the novelty claim.
- The evaluation is restricted solely to the 2-player Hanabi benchmark, offering no insight into whether this method is useful for spatially extended decision-making or cooperation in more complex environments.
- The paper focuses on comparisons to population-based methods and self-play but lacks critical comparisons to competitive ZSC baselines across all relevant categories: PBT (FCP, MEP, Comedei), self-play (E3T, CEC), and offline-to-online learning (GAMMA, GOAT, ROTATE).
- The human experiment results are limited in scale and lack detailed analysis beyond overall performance scores, preventing an understanding of the qualitative benefits or interaction mechanisms.
- While the method requires less online sampling, it is not clear whether the aggregate cost (offline data collection + offline training + online fine-tuning) ultimately outweighs the total sample efficiency of robust, fully online ZSC methods.

**Questions:**

- Given that concurrent work (e.g., GAMMA, GOAT) also leverages offline data to learn latent partner embeddings for ZSC, can the authors precisely articulate the novelty of their offline contribution beyond the use of clustering? A formal comparison of the information learned in the latent space would be beneficial.
- The paper lacks comparison against many competitive ZSC baselines. Please provide results comparing the proposed method against at least one representative baseline from the key missing categories: a Population-Based Training method (e.g., MEP or Trajedei) and a modern Zero-Shot Coordination method (e.g., E3T or CEC).
- As noted, the sample efficiency gain in the online phase is compelling, but the total cost is unclear. Could the authors provide an estimate or analysis of the relative computational cost (e.g., wall-clock time or GPU-hours) of the full pipeline (offline data collection + offline training + online fine-tuning) compared to training a state-of-the-art, fully online ZSC agent to the same performance level?
- Since the method is only evaluated in 2-player Hanabi, what is the authors' hypothesis, and supporting evidence (if any), regarding how the belief-guided counterfactual rollouts would scale or generalize to environments with spatially extended decision-making (e.g., StarCraft Multi-Agent Challenge) or larger team sizes?
- To strengthen the human-AI coordination claims, can the authors provide a more detailed analysis of the human experiments? Specifically, please report on the variance and statistical significance of the scores, and analyze whether the AI's learned conventions aligned qualitatively with observable human strategies.

---

### Official Review · Reviewer_6KHt · 2025-10-31

**Soundness:** 2
**Presentation:** 2
**Contribution:** 2
**Rating:** 2
**Confidence:** 5

**Summary:**

The paper proposes an offline-to-online framework for zero-shot coordination that first builds policy diversity from offline datasets and then performs belief-guided counterfactual fine-tuning online to adapt to unseen partners. Concretely, trajectories are embedded with a VAE and clustered into behavioral modes to train specialized agents and belief models, a best-response agent is learned against this pool, and online updates use counterfactual successor trajectories sampled under alternative teammate assumptions. Experiments on 2-player Hanabi and small-scale human–AI studies report improved cross-play performance and sample efficiency compared to offline and online baselines.

**Strengths:**

The paper focuses squarely on improving zero-shot coordination while reducing online interaction cost.

**Weaknesses:**

- Unclear motivation for offline-to-online. If the offline dataset is collected from humans (e.g., 200k Hanabi games), its cost likely dwarfs purely online training; if it is gathered with online self-play, it’s not obvious how this differs from simply doing online training in the first place. The paper does not clearly justify why an offline stage is necessary for ZSC, since the well-designed online population methods already provide many useful methods to ensure diversity.

- Limited experimental scope and external validity. All results are on Hanabi, which raises concerns about environment-specific tuning and limits claims about general applicability. The method’s scalability to other ZSC or mixed-motive domains (e.g., multi-player Hanabi variants, Overcooked, MeltingPot) is not demonstrated.

- Missing strong baselines and recent literature. The evaluation omits many post-2022 ZSC baselines, including methods that achieve strong zero-shot coordination without any human data (e.g., FCP-style or broader population-based approaches):

[1] Wang, Xihuai, et al. "Zsc-eval: An evaluation toolkit and benchmark for multi-agent zero-shot coordination." Advances in Neural Information Processing Systems 37 (2024): 47344-47377.

[2] Li, Yang, et al. "Cooperative open-ended learning framework for zero-shot coordination." International Conference on Machine Learning. PMLR, 2023.

[3] Zhao, Rui, et al. "Maximum entropy population-based training for zero-shot human-ai coordination." Proceedings of the AAAI Conference on Artificial Intelligence. Vol. 37. No. 5. 2023.

[4] Yu, Chao, et al. "Learning Zero-Shot Cooperation with Humans, Assuming Humans Are Biased." The Eleventh International Conference on Learning Representations.

[5] Yan, Xue, et al. "An efficient end-to-end training approach for zero-shot human-AI coordination." Advances in neural information processing systems 36 (2023): 2636-2658.

[6] Strouse, D. J., et al. "Collaborating with humans without human data." Advances in neural information processing systems 34 (2021): 14502-14515.

- Underpowered human-subject study. The human-AI evaluation includes only 10 participants, which is insufficient for statistical power and does not support strong claims about human generalization. The study lacks robust statistical analysis (e.g., power calculations, effect sizes, preregistered protocols), making the evidence largely anecdotal.

**Questions:**

- On Eq. (3) JSD regularizer appears to be an average of asymmetric KL terms \sum_j \mathrm{KL}(\pi^j \| \pi^i), not a Jensen–Shannon divergence, which is a factual mistake.

- Fig. 2 shows offline CP collapse, but your offline dataset is generated by OBL rather than real logged interactions. Why should this pathology generalize to real pre-collected data? Can you evaluate on human logs or mixed-policy logs to better approximate real “pre-existing data” scenarios?

- Why offline-to-online for ZSC? If the offline dataset were human-collected (e.g., 200k Hanabi games), its cost rivals or exceeds purely online training. If it’s collected via online rollouts, how is the offline stage substantively different from simply training online with population methods? What unique benefit does the offline stage provide for ZSC beyond initialization?

-  How do you quantify partner/policy diversity in the offline pool (e.g., convention coverage, action-entropy, cluster separation)? How does this diversity compare, under equal interaction budgets, to online population methods that explicitly maintain diverse teammates?

- What is the predictive accuracy/calibration of the belief over hidden information and over teammate types? Can you correlate belief error with CP performance and provide ablations where belief quality is degraded/improved?

---

### Official Review · Reviewer_DDeD · 2025-11-02

**Soundness:** 2
**Presentation:** 3
**Contribution:** 1
**Rating:** 2
**Confidence:** 4

**Summary:**

The paper introduces an offline-to-online framework for zero-shot coordination: it learns trajectory representations and clusters partners to form a diverse specialist pool, distills an offline best-response $Q_{\text{BR}}$, then performs belief-guided counterfactual rollouts for sample-efficient online adaptation. Evaluated on cooperative settings (e.g., 2-player Hanabi) and small-scale human–AI studies, the method improves cross-play performance and reduces online interaction cost relative to standard baselines. Contributions include a unified pipeline that couples offline diversity induction with belief-based fine-tuning, a diversity-regularized training objective over policies, and empirical evidence for efficient zero-shot coordination.

**Strengths:**

A key strength is the crisp, timely framing of the offline-to-online ZSC problem: initialize from diverse logs to train a competent $Q_{\text{BR}}$/$\pi$, then adapt online to unknown partners to improve $XP$ with limited interactions. This formulation clarifies the coverage–specialization trade-off and offers a practical path beyond overfitted $SP$ training.

**Weaknesses:**

- Factual inconsistency: the “JSD regularizer” is implemented as an average of directional $\mathrm{KL}(\pi_j\|\pi_i)$, not a symmetric $\mathrm{JSD}$, which can introduce asymmetry and instability unless carefully justified or replaced with a proper symmetric divergence.
- Motivation gap: the case for using offline data and, specifically, an offline-to-online pipeline is underdeveloped; the paper does not convincingly show why fully online adaptation or purely offline population methods are insufficient, nor provide ablations that isolate “offline only”, “online only”, and “offline→online” under matched budgets.
- Evaluation scope: key ZSC baselines (e.g., TrajeDi, FCP, Any-Play) are missing; the human–AI study has very small $N$ and unclear counterbalancing/statistics; sample-efficiency claims appear to ignore the cost of producing the offline dataset and report limited variance analysis.

**Questions:**

- Why is an offline stage necessary, and why specifically an offline-to-online pipeline? Please justify against “online-only” and “offline-only” alternatives with matched interaction/data budgets, and clarify under what conditions offline coverage helps rather than hurts XP due to convention bias.

---

### Note · Program_Chairs · 2026-01-17
**Submission Desk Rejected by Program Chairs**

The following references in this submission do not refer to real documents and/or have major errors in bibliographic information:


Ziqi Zhao et al. An efficient method for determining the optimal k in k-means clustering, 2024b.